# Primary Coenzyme Q10 Deficiency-Related Ataxias

**DOI:** 10.3390/jcm13082391

**Published:** 2024-04-19

**Authors:** Piervito Lopriore, Marco Vista, Alessandra Tessa, Martina Giuntini, Elena Caldarazzo Ienco, Michelangelo Mancuso, Gabriele Siciliano, Filippo Maria Santorelli, Daniele Orsucci

**Affiliations:** 1Unit of Neurology, San Luca Hospital, Via Lippi-Francesconi, 55100 Lucca, Italy; piervito.lopriore@gmail.com (P.L.); marco.vista@uslnordovest.toscana.it (M.V.); martina.giuntini@uslnordovest.toscana.it (M.G.); elenacaldarazzoienco@gmail.com (E.C.I.); 2Neurological Institute, Department of Clinical and Experimental Medicine, University of Pisa, 56126 Pisa, Italy; michelangelo.mancuso@unipi.it (M.M.); gabriele.siciliano@unipi.it (G.S.); 3Molecular Medicine, IRCCS Stella Maris Foundation, 56122 Pisa, Italy; aletessa@gmail.com (A.T.); filippo.santorelli@fsm.unipi.it (F.M.S.)

**Keywords:** ataxia, cerebellum, coenzyme Q10, mitochondrial diseases, primary coenzyme Q10 deficiencies

## Abstract

Cerebellar ataxia is a neurological syndrome characterized by the imbalance (e.g., truncal ataxia, gait ataxia) and incoordination of limbs while executing a task (dysmetria), caused by the dysfunction of the cerebellum or its connections. It is frequently associated with other signs of cerebellar dysfunction, including abnormal eye movements, dysmetria, kinetic tremor, dysarthria, and/or dysphagia. Among the so-termed mitochondrial ataxias, variants in genes encoding steps of the coenzyme Q10 biosynthetic pathway represent a common cause of autosomal recessive primary coenzyme Q10 deficiencies (PCoQD)s. PCoQD is a potentially treatable condition; therefore, a correct and timely diagnosis is essential. After a brief presentation of the illustrative case of an Italian woman with this condition (due to a novel homozygous nonsense mutation in *COQ8A*), this article will review ataxias due to PCoQD.

## 1. Introduction

Primary mitochondrial diseases (PMDs) are an extremely complex group of rare neurological and/or multisystem diseases caused by the impairment of the respiratory chain, or electron transport chain, site of oxidative phosphorylation [1]. A particularly rare subgroup of PMDs is primary coenzyme Q10 deficiency (PCoQD), also called primary ubiquinone deficiency, due to recessive mutations in at least 10 genes involved in the molecular pathways required for the biosynthesis of this cofactor. Coenzyme Q10 deficiency has also been identified as a secondary consequence of other conditions [2] not discussed here.

Coenzyme Q10 (or ubiquinone), a key component of the electron transport chain, is an endogenously synthesized molecule, mostly localized in the mitochondria. In mammalian cells, at least 14 proteins are needed for the synthesis of coenzyme Q10, and its biosynthesis starts with the formation of (I) a 4-hydroxybenzoic acid derived from tyrosine or phenylalanine and (II) a polyisoprenoid tail derived from the mevalonate pathway [2]. Different enzymes, encoded by *COQ1*, *COQ2*, *COQ3*, *COQ5*, *COQ6*, and *COQ7*, catalyze the formation of coenzyme Q10. Moreover, *COQ4*, *COQ8A*, *COQ8B*, and *COQ9* have regulatory functions over the other biosynthetic proteins [2]. 

Coenzyme Q10 is composed of a benzoquinone ring, which confers the redox properties of the molecule, and a polyprenoid tail responsible for its lipophilicity. The benzoquinone ring may exist in three redox states: fully oxidized (ubiquinone), semiquinone radical (ubisemiquinone), and fully reduced (ubiquinol) [2]. The redox capacity of coenzyme Q10 allows this molecule to directly reduce reactive oxygen species (ROS) and to regenerate other antioxidants, e.g., tocopherol and ascorbate. However, the principal metabolic role of coenzyme Q10 takes place in the mitochondrial oxidative phosphorylation system. Here, there are two pools of coenzyme Q10 molecules: (1) a pool attached in the super-complexes I + III, exclusively dedicated to the oxidation of NADH; and (2) a free pool available for complex II (for oxidation of FADH2) and any other enzyme that uses coenzyme Q10 as a cofactor.

Moreover, coenzyme Q10 is involved in other cellular pathways besides oxidative phosphorylation, i.e., it is a cofactor of several enzymes (mostly FAD-linked oxidoreductases) involved in lipid and amino acids, nucleotides metabolism, and sulfide detoxification, and it contributes to the cellular redox control and programmed cell death modulation [3].

Cerebellar ataxia is one of the phenotypes that have been described in PCoQD [4]. Ataxia is a neurological syndrome characterized by the imbalance (e.g., truncal ataxia, gait ataxia) and incoordination of limbs while executing a task (dysmetria), caused by the dysfunction of the cerebellum or its connections. It is frequently associated with other signs of cerebellar dysfunction, including abnormal eye movements, dysmetria, kinetic tremor, dysarthria, dysphagia, and possibly cognitive impairment [5]. In PCoQD, cerebellar ataxia is typically associated with neuroimaging findings of cerebellar atrophy [1].

Traditionally, clinical manifestations of PCoQD have been classified into five discrete phenotypes: encephalomyopathy (with recurrent myoglobinuria, encephalopathy, and mitochondrial myopathy); cerebellar ataxia; severe infantile multisystemic form; isolated myopathy (with muscle weakness, myoglobinuria, exercise intolerance, and elevated CK levels); and steroid-resistant nephrotic syndrome [3]. However, this classification is outdated since the PCoQD phenotypic spectrum is much wider, and overlapping phenotypes have been described [6].

In this narrative review, after a brief presentation of the illustrative case of an Italian woman with cerebellar ataxia due to a novel homozygous stop mutation in *COQ8A*, we focus our attention on mitochondrial ataxias due to PCoQD and on their treatment options.

## 2. Material and Methods

This study was conducted with a focused literature review on ataxia manifestation in PCoQD. The Pubmed database was utilized, at the end of February 2024, as the source for the literature search. The following specific Medical Subject Headings (MeSH) terms were used: “coenzyme Q10” and “ataxia”. A total of 262 abstracts in English were reviewed to identify relevant publications (see References). Additional articles were also identified by reviewing the bibliographies of relevant studies. Review articles, original articles, cohort studies, case series, and case reports were included. The search was limited to articles published in English, with no restrictions on the publication date.

## 3. Case Report

The patient briefly presented in Figure 1 had a slowly progressive cerebellar syndrome since she was a child. Neuroimaging only showed diffuse cerebral and cerebellar atrophy.

This woman, from a non-consanguineous Italian family, had gait and balance disturbances since she was a child (with repeated falls since then). Her family history was unremarkable. Neurological examination revealed marked gait and limb ataxia and mild dysmetria and cerebellar dysarthria. Her SARA (scale for the assessment and rating of ataxia) score was 18. The fiberoptic Evaluation of Swallowing was unremarkable. Magnetic Resonance Imaging (MRI) only showed diffuse cerebral and cerebellar atrophy (see Figure 1). There was not any history of vascular disorders, alcohol abuse, or toxins exposure. A complete work-up excluded common causes of cerebellar ataxia, including vitamin deficiencies and autoimmune conditions. 

Next-generation sequencing (NGS) studies led to the identification of the homozygous c.547C>T (p.Q183*) stop mutation in *CoQ8A* (confirmed by Sanger sequencing). This variant was classified as “pathogenic” by the ACMG criteria, with a CADD score of 41. Muscle biopsy was not performed because of ethical concerns in an already-diagnosed patient. For this reason, coenzyme Q10 levels were not measured in the muscular tissue, and serum coenzyme Q10 determination is usually not considered useful. She was prescribed Coenzyme Q10 600 mg/day. After two years of such therapy, her neuromotor examination was unchanged, but she also developed mild cognitive impairment (MCI), with a Mini Mental State Examination (MMSE) score of 22/30, when reevaluated at age 79.

In conclusion, after the identification of the novel homozygous nonsense variant in *CoQ8A*, she was prescribed high-dosage coenzyme Q10, even if it was very likely an optimal response to this treatment might require a much earlier start, ideally before the development of cerebellar atrophy. After two years of such therapy, she remained clinically stable and her neuromotor examination was unchanged. A timely diagnosis (which was not possible for our elderly patient) would be crucial.

## 4. Primary Coenzyme Q10 Deficiency-Related Ataxia: The Example of *CoQ8A*

In PCoQD, manifestations of central nervous system involvement include encephalopathy, epilepsy, dystonia, spasticity, intellectual disability, and cerebellar ataxia. With regard to the peripheral nervous system, distal motor neuropathy has been reported in some cases [6]. Cerebellar ataxia, variably associated with other neurological and systemic symptoms, is one of the most common clinical presentations of PCoQD. The most frequent genetic causes of PCoQD-related cerebellar ataxia are pathogenic variants in *COQ8A* (autosomal-recessive cerebellar ataxia type 2 -ARCA2- or SCAR9, OMIM code: 612016).

The *COQ8A* gene (gene ID: 56997), also known as *ADCK3*, is located on Chromosome 1 and comprises 15 exons. The *COQ8A* gene encodes for an enzyme belonging to the UbiB protein kinase-like family, classified as an atypical (or non-canonical) kinase. COQ8A localizes to the matrix face of the inner mitochondrial membrane, interacting, together with COQ8B, with complex Q, the core enzyme machinery made up of the COQ3, COQ5, COQ6, and COQ7 proteins, which is involved in coenzyme Q10 biosynthesis [7,8]. Specifically, COQ8A has ATPase and small-molecule kinase activity, which support complex Q interactions and assembly, and phosphorylates complex Q components, ensuring its stability and activity, respectively [7,8]. The protein kinase-like (PKL) domain of the COQ8A protein, as for the other UbiB family proteins, possesses an N lobe insert (NLI) and an N-terminal extension (NTE) with specific motifs, illustrated in Figure 2 [8]. UbiB family proteins also have an atypical alanine-rich loop (AAAS motif) replacing the canonical glycine-rich loop in the nucleotide binding domain, an ExD motif in the NLI, and a modified catalytic loop (Figure 2) [8]. It has been shown that the AAAs motif in the nucleotide-binding loop determines coenzyme selectivity and inhibits COQ8A autophosphorylation [8]. The KxGQ motif in the NTE domain plays a central role in inhibiting cis autophosphorylation but also supports COQ8A ATPase activity and small-molecule kinase activity (e.g., a CoQ intermediate or cofactor), as suggested by the “unorthodox PKL model” [7].

Constitutive *COQ8A* knockout (*Coq8a*^−/−^) mice develop a mild phenotype characterized by progressive ataxia, seizures, and exercise intolerance driven by complex Q instability and coenzyme Q10 deficiency. Cerebellar Purkinje neurons show altered electrophysiological function and dark cell degeneration. Therefore, this mouse model recapitulates the cardinal phenotypic features of ARCA2 [7]. Recently, a Purkinje-specific conditional *COQ8A* knockout mouse has been generated, demonstrating the causative role of Purkinje degeneration in PCoQD-related ataxia. In this model, Purkinje neurons show specific complex IV alteration (during the pre-symptomatic stages), abnormal dendritic arborizations, altered mitochondria functioning and intracellular calcium dysregulation, which can be rescued by coenzyme Q10 supplementation [9].

More than 120 *COQ8A*-related PCoQD patients and 75 pathogenic variants (including missense and loss-of-function) in *COQ8A* have been reported so far. *COQ8A*-ataxia is inherited in an autosomal recessive manner, and the first case was reported in 2008 [10]. The pathogenic and likely pathogenic variants in the *COQ8A* gene associated with an ataxic phenotype are illustrated in Figure 2.

**Figure 2 jcm-13-02391-f002:**
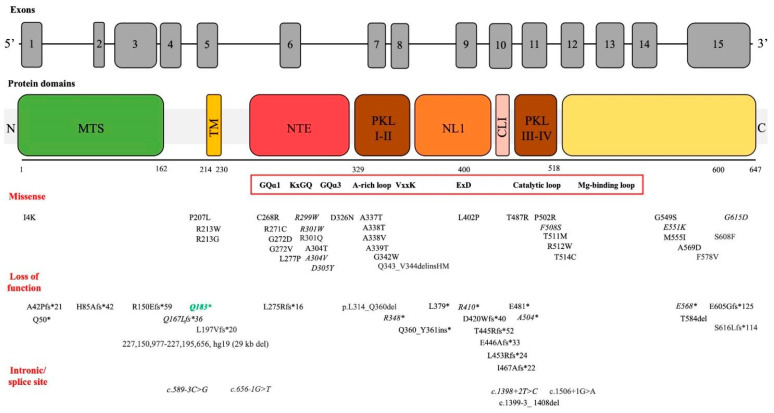
Variants associated with ataxia across COQ8A protein domains. Graphical overview of variants in relation to the COQ8A protein domains and functional motifs (in the red square). The vast majority of variants came from Traschütz et al.’s [11] clinico-genetical multicenter study. Moreover, we searched pathogenic and likely pathogenic variants reported in association with ataxia phenotypes in the ClinVar Miner database (https://clinvarminer.genetics.utah.edu/ accessed on 7 March 2024). Variants reported in homozygosity are reported in italics. The novel homozygous variant identified in our patient is highlighted in green. N: N-terminus; MTS: mitochondrial targeting sequence; TM: transmembrane domain; PKL: subdomains of the protein kinase-like superfamily; NTE: UbiB family-specific N-terminal extension; NLI: N lobe insert; CLI: C lobe insert. KxGQ motif (residue 276–279); AAAS motif (residue 337–340).

In the largest multicenter study describing the clinico-genetic features of *COQ8A*-ataxia, missense variants were found to be predominantly grouped around the active site of the COQ8A protein (protein kinase-like superfamily domains) [11]. Whereas loss-of-function variants led to the termination of the protein, missense variants were predicted to cause steric and electrostatic clashes, lost interactions between amino acids, mitochondrial targeting impairment, and the disruption of the transmembrane or GQα helix [11].

According to this study, some genotype–phenotype correlations were found: biallelic loss-of-function variants primarily presented with phenotypes limited to the cerebellar ataxia; biallelic missense variants were associated with a multisystemic phenotype (cognitive impairment, epilepsy, myoclonus, dystonia, myopathic features); biallelic missense variants within the AAAS motif presented with ataxia, generally without epilepsy; and variants within the KxGQ domain presented with multisystem phenotypes (developmental delay, epilepsy, and pyramidal signs) [11]. Regarding missense variants, we may speculate that these genotype–phenotype correlations reflect the functional difference in the AAAs motif and KxGQ domain, the former having a role in inhibiting protein autophosphorylation and the latter being essential for the COQ8A role in complex Q stabilization through unorthodox PKL activity [7,8].

The *COQ8A*-related ataxia shows a wide spectrum of clinical presentations even in the same family [11,12]. In about half of the cases, the age of onset is before 6 years, with the majority of patients being affected by age 15 [11]. Onset in adulthood has been reported [13]. Cerebellar dysfunction, universally present in the two biggest *COQ8A*-related PCoQD cohorts, typically presents as gait ataxia and a loss in balance [11,14]. During the disease progression, other cerebellar signs such as limb dysmetria, dysdiadochokinesia, abnormal eye movements, intentional tremor, hypotonia, and dysarthria appear [11,14]. Cognitive impairment and intellectual disabilities are present in about half of cases, with some patients manifesting psychomotor regression and about 25% patients showing neuropsychiatric symptoms (depression, anxiety, psychosis, aggressivity) [11,14]. Epilepsy (from well-controlled to multidrug-resistant forms) and myopathic features (especially exercise intolerance) are present in about 25% of patients. Additional manifestations may include movement disorders (cervical dystonia or upper limb focal dystonia, chorea, myoclonus, head and postural tremors), eye movement alterations such as saccadic ocular pursuit, external ophthalmoplegia, and oculomotor apraxia, and, more rarely, pyramidal signs and dysphagia [11,14]. Interestingly focal dystonia, even preceding the ataxia phenotype, could be the presenting symptoms of *COQ8A*-related PCoQD. Moreover, ataxia and dystonia represent a separate phenotype cluster [11,15]. Of note, other typical “mitochondrial” red flags such as stroke-like episodes, hearing loss, optic atrophy, pigmentary retinopathy, cataract, diabetes, cardiomyopathy, and renal and liver dysfunction have been reported only rarely [11]. Overall, the disease progression is usually mild to moderate, with rare patients experiencing worse disease trajectories with death before adolescence [11]. The combination of cross-sectional (*n* = 34) and longitudinal (*n* = 7) assessments also indicated the slow progression of ataxia (SARA: 0.45/year) [11].

Cerebellar atrophy, sometimes limited only to the vermis, is the predominant neuroimaging feature of *COQ8A*-related ataxia. Cerebellar involvement can be evident in the anterior and posterior lobes, superior cerebellar peduncle, and pontine crossing tracts. T2 hyperintensities in dentate nuclei and dorsal pons have been reported. Moreover, in about 25% of patients, supratentorial (frontal, insular, and parietal) atrophy has also been observed [11].

## 5. Other Causes of Primary Coenzyme Q10 Deficiency-Related Ataxia

The *COQ4* gene (gene ID: 51117) is located in Chromosome 9 and comprises seven exons. The *COQ4* gene encodes for a protein which was previously proposed to fulfill a structural role in the organization of complex Q. Recent evidence suggests that the COQ4 enzyme acts as an oxidative decarboxylase, substituting in a single step the carboxylic acid group with a hydroxyl group on carbon 1 of coenzyme Q10 precursors [16]. More than 40 *COQ4*-related PCoQD patients and 25 pathogenic variants in the *COQ4* gene have been identified so far. *COQ4A*-related PCoQD is inherited in an autosomal recessive manner [17]. Compound heterozygous or homozygous pathogenic variants in *COQ4* typically cause childhood-onset neurodegeneration, with phenotypes ranging from severe early-onset epileptic encephalopathy and brain anomalies and less severe encephalopathy with stroke-like episodes to a moderate phenotype with a stable disease course in the absence of brain anomalies; cardiomyopathy and severe multi-system phenotypes have been described too [18,19]. Cerebellar degeneration has been observed in infants with severe encephalopathy harboring bi-allelic variants in *COQ4*, and peculiar *COQ4*-related ataxia phenotypes have been reported [19]. Two siblings with childhood-onset spinocerebellar syndrome and stroke-like episodes have also been described [20]. Further childhood-onset ataxia phenotypes, in variable association with motor impairment (spasticity, wheelchair-dependency in late childhood), cognitive impairment (neurodevelopmental disorder), and seizures, have been reported [21,22]. Moreover, six cases of adult-onset ataxia-spasticity spectrum phenotypes were recently reported [23].

The *COQ2* gene (gene ID: 27235) is located on Chromosome 4 and comprises seven exons. The *COQ2* gene encodes for p-hydroxybenzoate:polyprenyl transferase, which catalyzes the second step of the final reaction sequence of coenzyme Q10 biosynthesis, the condensation of 4-hydroxybenzoate with polyprenyl-pyrophosphate [24]. The first case of *COQ2*-related PCoQD was reported in 2006 and was the first molecular defect responsible for PCoQD to be identified [25]. *COQ2* mutations are associated with a wide clinical spectrum, ranging from fatal neonatal multisystemic disease to late-onset encephalopathy, including steroid-resistant nephrotic syndrome. A clear genotype–phenotype correlation, based on residual coenzyme Q10 production of the mutant alleles, has been established [26]. *COQ2* variants have been associated with a syndrome resembling multiple-system atrophy (MSA), a non-monogenic neurodegenerative alpha synucleinopathy disorder characterized by autonomic failure in addition to various combinations of parkinsonism, cerebellar ataxia, and pyramidal dysfunction. Particularly, homozygous mutations (p.Met128Val-p.Val393Ala/p.Met128Val-p.Val393Ala) and compound heterozygous mutations (p.Arg387*/p.Val393Ala) have been observed in MSA-like families. Moreover, common variants in the Japanese population (p.Val393Ala and p.Leu25Val) and multiple rare variants have been associated with sporadic cases, particularly with a cerebellar MSA phenotype [27,28,29]. Interestingly, decreased coenzyme Q10 levels have also been observed in various tissues of MSA patients not carrying the p.Val393Ala variant [30]. The exact mechanisms linking MSA and Coenzyme Q10 deficiency must still be clarified.

The *COQ5* gene (gene ID: 84274) is located on Chromosome 12 and comprises eight exons. The *COQ5* gene encodes for the enzyme responsible for the only C-methylation step in coenzyme Q10 biosynthesis. To date, a single report of *COQ5*-related isolated PCoQD has been presented. In 2017, three female siblings from a Middle Eastern Iraqi-Jewish descent family presented with early-onset cerebellar ataxia and encephalopathy characterized by cognitive disability and myoclonic-epilepsy, resulting from a biallelic duplication of 9590bp in *COQ5* [31].

All the likely pathogenic or pathogenic variants in the *COQ2*, *COQ4*, and *COQ5* genes associated with the ataxia phenotype are reported in Table 1.

## 6. Treatment Options

PCoQDs are generally considered among the few primary mitochondrial disorders for which a treatment option is available [4,6]. High-dose oral coenzyme Q10 supplementation (ranging from 5 to 50 mg/kg/day) can limit disease progression and reverse some manifestation, especially when the condition is recognized sufficiently early, and no severe renal or neurological damage is established [32]. Soluble formulations are generally preferred, even if the bioavailability of orally supplemented ubiquinone is considered inherently low [6,33]. Diarrhea and anorexia were found as common side effects of ubiquinone supplementation therapy [4,6].

The clinical response is highly variable and depends on both the genotype and disease burden, with some authors suggesting that the evidence for the efficacy of the treatment is actually weak [34,35]. There are some reports of symptom worsening, including ataxia and cognitive functioning, after coenzyme Q10 supplementation withdrawal [12,31,36].

*COQ8A*-related ataxia shows the most consistent data, also given the higher number of cases reported. Ataxia itself, among a constellation of various symptoms, is one of the most responsive features [37]. Few cases of a dramatic improvement in ataxia after ubiquinone treatment, even with a total withdrawal of cerebellar signs and a restoration of unaided walk ability, were reported [36,38]. Moreover, improvements in additional neurological manifestations such as dystonia, epilepsy, myoclonus, cognitive function, and muscle weakness/exercise intolerance have been reported [11,37,38]. Despite these positive preliminary observations, approximately 50% of patients do not respond to coenzyme Q10 supplementation, with variable therapy effectiveness between responders [11]. No reliable predictor of therapeutic efficacy in *COQ8A*-related ataxia is available [37]. A multicenter study showed that the age of onset, disease duration at treatment, disease severity, total coenzyme Q10 daily dose, genotype, and imaging and laboratory parameters (such as the CoQ10 level in the muscle) do not predict the treatment response [11]. A recent work suggested that metabolic ^31^phosphorus magnetic resonance spectroscopy imaging may map the pretreatment status of *COQ8A*-patients, assessing the cerebellar bioenergetic status and predicting the treatment response, but further studies are needed [39]. The coenzyme Q10 supplementation duration is not specified. A longitudinal assessment showed a reduction of −0.88 points/year in the SARA scale [11]. A 2-year follow-up study demonstrated a significant reduction in the international cooperative ataxia rating scale (ICARS), especially in posture and kinetic function scores [40]. Another study reported a possibly better effect of coenzyme Q10 supplementation in patients treated for 1 year than in patients treated for only 6 months [41].

Regarding *COQ4*-related ataxia, partial responses have been observed. One case showed neuromuscular and motor skills improvement after coenzyme Q10 supplementation [22]. Another paper reported a significant improvement in the ataxia phenotype, especially in gait and stance parameters quantified with the SARA scale, in two affected siblings following one month of coenzyme Q10 supplementation. The treatment was subsequently continued for one year, resulting in substantial stability [21].

A three-year follow-up study with high-dose coenzyme Q10 supplementation (1200 mg/day) in a patient with an advanced stage MSA-like phenotype, harboring compound heterozygous *COQ2* mutations (p.Arg387*/p.Val393Ala), showed clinical stability, together with increased total coenzyme Q10 levels in cerebrospinal fluid and plasma after supplementation, as well as brain oxidative metabolism measured with ^15^O_2_ positron emission tomography [42]. High-dose ubiquinol in MSA patients is well-tolerated and led to a significantly smaller decline in the MSA-specific scale (i.e., Unified Multiple System Atrophy Rating Scale—part 2) in a recent multicenter, randomized, double-blinded, placebo-controlled phase 2 trial [43]. 

Regarding *COQ5*-related ataxia, in three cases, oral coenzyme Q10 supplementation (15 mg/kg/day) led to a significant decrease in the ICARS score [31].

Interestingly, coenzyme Q10 supplementation has been tested in a spinocerebellar ataxias (SCAs) cohort and has been associated with better clinical outcomes in SCA1 and SCA3; however, it did not appear to influence clinical progression within 2 years [44].

## 7. Non-Pharmacological Management

Patient care standards for PMDs, including PCoQD, are better detailed elsewhere [45]. Appropriate care standards also include enteral nutrition and/or assisted ventilation in subjects with advanced disease. Furthermore, these patients should be offered age-appropriate vaccination [46].

With specific regard to ataxia, its treatment remains difficult [47]. The mainstays of the treatment of cerebellar ataxias are currently physiotherapy, occupational therapy, and speech (and swallowing) therapy. Even if evidence-based guidelines for the physiotherapy of degenerative cerebellar ataxia need to be developed, preliminary data show that coordinative training improves motor function in both adult and juvenile patients with cerebellar disease [47]. The benefits of intensive whole-body coordinative training on balance and mobility function in degenerative cerebellar disease have been demonstrated [48]. As for coenzyme Q10 supplementation, we can intuitively hypothesize that a rehabilitative program should be started as soon as possible, ideally before the development of cerebellar atrophy.

## 8. Discussion and Conclusions

Cerebellar ataxia is one of the phenotypes that have been described in PCoQD. To date, mutations in four different genes belonging to the coenzyme Q10 biosynthetic pathway have been described in PCoQD-related ataxia phenotypes, with *COQ8A* mutation clearly representing the vast majority (Figure 3).

Cerebellar ataxia manifestation in PCoQD is typically characterized by infantile or childhood onset, whereas adult-onset ataxia has been less frequently reported. It can be present in isolation or in association with dystonia, especially for *COQ8A* variants. However, in most cases, it is associated with other neurological manifestations such as epilepsy, encephalopathy, spasticity, dystonia, neurodevelopmental disorder, cognitive impairment, myopathy, or exercise intolerance both for *COQ8A*- and *COQ4*-related disease. A progressive, childhood-onset spastic-ataxia phenotype associated with mild cognitive impairment has been reported in patients harboring *COQ4* mutations, whereas *COQ2* mutation has been linked to MSA-like phenotypes. To date, a single report of *COQ5*-related isolated PCoQD has been reported. It is important to underline that mutations in *COQ4* and *COQ2* have been linked to other different PCoQD phenotypes, such as severe infantile multisystemic form, retinopathy, hypertrophic cardiomyopathy, and steroid-resistant nephrotic syndrome.

The metabolism and functions of coenzyme Q10 are more complex than usually thought (see Figure 3). Further studies are strongly needed to unravel such complexity and to explain, for instance, why mutations in some enzymes in the pathway show constant association with cerebellar ataxia, whereas others do not.

PCoQD-related ataxia diagnosis is principally driven by molecular genetic testing. Coenzyme Q10 level measurement, traditionally used to diagnose PCoQD, is now limited to confirm genetic testing results or to explore suspected cases in which molecular testing fails (i.e., the presence of a variant of unknown significance in COQ genes). Regarding biochemical testing, the main evidence of PCoQD is: (1) reduced CoQ10 levels in skeletal muscle and (2) decreased activities of complex I + III and II + III in frozen muscle samples [48,49,50]. Alternatively, coenzyme Q10 level measurement can be performed in skin fibroblast or blood mononuclear cells, whereas the plasma level is not useful because it reflects dietary intake [48,49]. Furthermore, there are cases of PCoQD with normal muscle and fibroblast coenzyme Q10 levels, as reported by Mero et al. [22].

In conclusion, it is important to consider PCoQD in patients with undiagnosed ataxia. Supplementation with coenzyme Q10 (at least 300–600 mg per day) may stabilize the disease and/or improve symptoms [46]. However, the response to Q10 supplementation may be incomplete or ineffective, and large studies are needed to dissect this phenomenon. Meanwhile, a timely diagnosis is of fundamental value since it promotes early coenzyme Q10 supplementation and thus a potential benefit for the patient. NGS, and especially exome studies given the high phenotype and genotype heterogeneity, can help to achieve this goal.

## Figures and Tables

**Figure 1 jcm-13-02391-f001:**
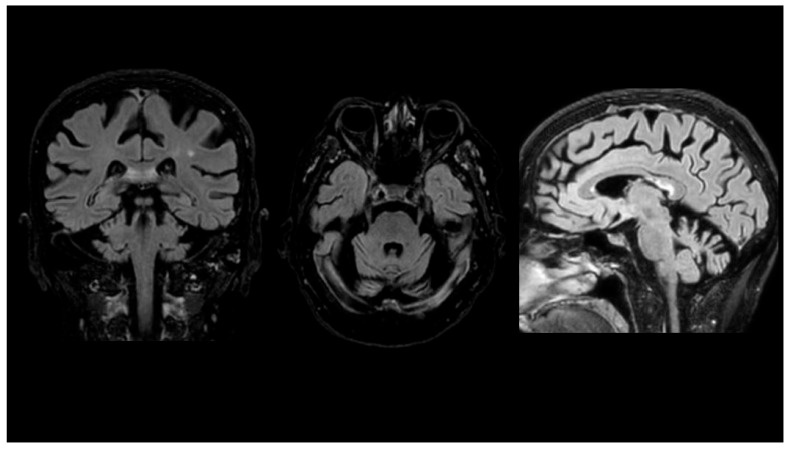
Cerebral and cerebellar atrophy in a 77-year-old woman with primary coenzyme Q10 deficiency (MRI-FLAIR). Targeted sequencing studies led to the identification of the homozygous c.547C>T (p.Q183*) stop mutation in *COQ8A* (confirmed by Sanger sequencing).

**Figure 3 jcm-13-02391-f003:**
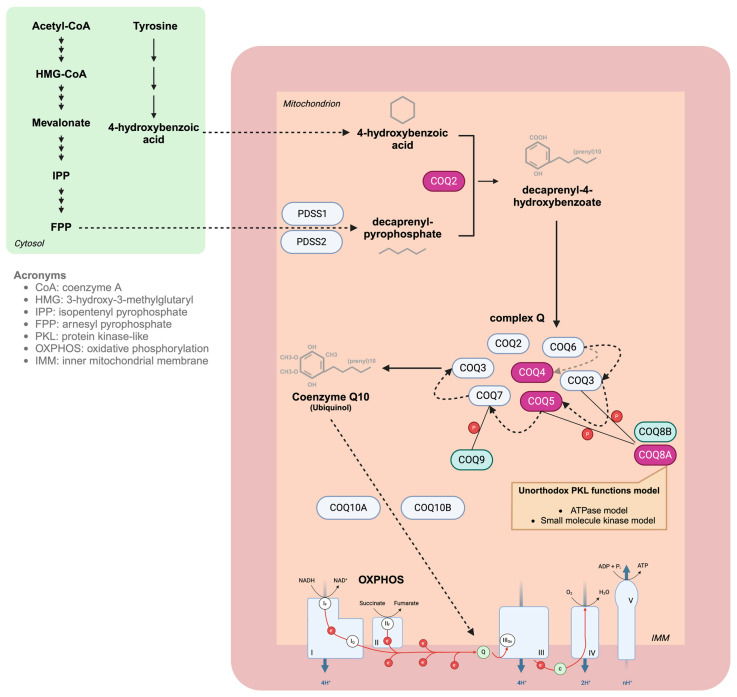
Graphical illustration of Coenzyme Q10 Biosynthesis and mitochondrial oxidative phosphorylation. Genes associated with PCoQD-ataxia phenotypes are highlighted in violet. PDSS1-2: decaprenyl diphosphate synthase subunit 1 and 2; COQ2: coenzyme Q2, polyprenyltransferase; COQ3: coenzyme Q3, methyltransferase; COQ5: coenzyme Q5, methyltransferase; COQ6: coenzyme Q6, monooxygenase; COQ7: coenzyme Q7, hydroxylase.

**Table 1 jcm-13-02391-t001:** Variants associated with PCoQD-related ataxia. List of PCoQD-related ataxia variants (excluding *COQ8A*) and related phenotypes. Het: heterozygous; Hom: homozygous; P: pathogenic; LP: likely pathogenic.

Gene	DNA Variant	Protein Variant	Zygosity	Mutation Type	Phenotype	References
*COQ4*	c.190C>T	p.Pro64Ser	Homozygous	Missense	Infantile-onset spastic ataxia, epilepsy	[19]
*COQ4*	c.230C>T	p.Thr77Ile	Homozygous	Missense	Childhood-onset spinocerebellar ataxia with stroke-like episodes	[20]
*COQ4*	c.164G>T	p.Gly55Val	Homozygous	Missense	Childhood-onset ataxia with partial epilepsy and cognitive impairment	[21]
*COQ4*	c.305G>A	p.Arg102His	Compound het in association with c.284G>A	Missense	Childhood-onset ataxia, neurodevelopmental disorder	[22]
*COQ4*	c.284G>A	p.Gly95Asp	Compound het in association with c.305G>A	Missense	Childhood-onset ataxia, neurodevelopmental disorder	[22]
*COQ4*	c.577C>T	p.Pro193Ser	Compound het in association with c.718C>T	Missense	Childhood onset ataxia, progressive spasticity	[22]
*COQ4*	c.718C>T	p.Arg240Cys	Compound het in association with c.577C>T	Missense	Childhood-onset ataxia, progressive spasticity	[22]
*COQ4*	c.305G>A	p.Arg102His	Compound het in association with c.473G>A	Missense	Childhood-onset spastic ataxia, mild cognitive impairment	[23]
*COQ4*	c.473G>A	p.Arg158Gln	Compound het in association with c.305G>A	Missense	Childhood-onset spastic ataxia, mild cognitive impairment	[23]
*COQ4*	c.434G>A	p.Arg145His	Compound het in association with c.437T>G	Missense	Childhood-onset spastic ataxia, postural tremor	[23]
*COQ4*	c.437T>G	p.Phe146Cys	Compound het in association with c.434G>A	Missense	Childhood-onset spastic ataxia, postural tremor	[23]
*COQ4*	c.202+4A>C	-	Homozygous	Intronic	Childhood-onset ataxia, mild cognitive impairment	[23]
*COQ2*	c.382A>G	p.Met128Val	Homozygous in association with hom c.1178T>C	Missense	MSA-p (definite), cerebellar signs, retinitis pigmentosa	[27]
*COQ2*	c.1178T>C	p.Val393Ala	Homozygous in association with hom c.382A>G	Missense	MSA-p (definite), cerebellar signs, retinitis pigmentosa	[27]
*COQ2*	c.1159C>T	p.Arg337Ter	Compound het in association with c.1178T>C	Nonsense	MSA-c (probable)	[27]
*COQ2*	c.1178T>C	p.Val393Ala	Compound het in association with c.1159C>T	Missense	MSA-c (probable)	[27]
*COQ5*	[Chr12(GRCh37):120940098-120949687]	-	Homozygous	Tandem duplication (9590 bp)	Childhood-onset cerebellar ataxia, encephalopathy, generalized tonic-clonic seizure, developmental delay	[31]

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
