# Peer review of "Primary Coenzyme Q10 Deficiency-Related Ataxias"

_jcm, 2024, doi:10.3390/jcm13082391_

Round 1
Reviewer 1 Report
Comments and Suggestions for Authors
This is an article addressing primary coenzyme Q10 deficiencies related clinical manifestations mostly ataxias which might help Neurologist and other physicians.
The manuscript is well-written but lacks details regarding the methods of literature search, the type of review and PRISMA statement. I would also recommend to organize the manuscript to introduction, methods, results, discussion, and conclusion.
I would recommend to provide complete history of the patient including risk factors/ causes for cerebellar ataxia: vascular disorders, history of alcohol use, medications, and toxins exposure. Authors did not explain whether they performed complete work up to rule out other causes for cerebellar ataxia work up: complete blood picture, thiamine level, HbA1C, TSH, B12, Vitamin E, ANA and GAD 65 antibodies. If they are negative, would recommend to say work up was negative. I would recommend to compare pre and post treatment MRI brain imaging.
This review article is not unique but authors made a good summary of neurological manifestation due to coenzyme Q10 deficiencies.
The conclusions are consistent with evidence cited in the article. These suggestions are from a Neurologist standpoint.
Comments on the Quality of English Language
This manuscript is well written but needs minor editing.
Author Response
We are very grateful for these suggestions.
In answer to the specific points:
- This is an article addressing primary coenzyme Q10 deficiencies related clinical manifestations mostly ataxias which might help Neurologist and other physicians. The manuscript is well-written but lacks details regarding the methods of literature search, the type of review and PRISMA statement. I would also recommend to organize the manuscript to introduction, methods, results, discussion, and conclusion. --> we reorganized the manuscript as requested, including a detailed "methods" section. We have specified that this is a narrative review (PRISMA statement not appliable)
I would recommend to provide complete history of the patient including risk factors/ causes for cerebellar ataxia: vascular disorders, history of alcohol use, medications, and toxins exposure. Authors did not explain whether they performed complete work up to rule out other causes for cerebellar ataxia work up: complete blood picture, thiamine level, HbA1C, TSH, B12, Vitamin E, ANA and GAD 65 antibodies. If they are negative, would recommend to say work up was negative. I would recommend to compare pre and post treatment MRI brain imaging. --> we reorganized the manuscript as requested, including a detailed "case report" section, including the requested information.
This review article is not unique but authors made a good summary of neurological manifestation due to coenzyme Q10 deficiencies. The conclusions are consistent with evidence cited in the article. These suggestions are from a Neurologist standpoint. --> We thank the reviewer for the globally positive comments and the very useful suggestions. Every change in the text is highlihted.
Reviewer 2 Report
Comments and Suggestions for Authors
In this review, Lopriore et al., introduced primary coenzyme Q10 deficiency-related ataxias. The authors examined the mutations in genes encoding various enzymes necessary in the synthesis of coenzyme Q10 and their relationship to clinical cases. The authors ended the review with an overview of current treatment plans and their efficacy and a brief summary of non-pharmacological management. The review is well-written, organized, and easy to follow. However, there are places where the authors should revise to increase the clarity.
1. In Figure 2, 15 exons are shown for COQ8A gene. However, in line 106-107, the authors mentioned that the COQ8A gene has 18 exons. Please clarify.
2. The authors should explain the biochemical and molecular functions of the motifs listed in Figure 2, such as KxGQ, AAAS etc. This is particularly important for the discussion in line 145-150. The authors should also give some discussions on the speculations or evidence on the differential clinical severity of different missense mutations.
3. In discussing the mutations in COQ4, COQ2 and COQ5, the authors should compare the clinical severity among them as well as to the severity of COQ8A? Are different mutations associated with different physiological levels of Coenzyme Q10? Could the authors provide a brief illustrative figure on the biosynthetic pathway for coenzyme Q10 and how these various enzymes fit in the pathway? The authors then could have some discussions to synthesize the relationship between various Coenzyme Q10 related mutations and clinical severity.
4. The authors mentioned in the introduction as well as in conclusion that early diagnosis is key because it promotes early coenzyme Q10 supplementation. However, in line 255-257, the authors used reference [10] to suggest that disease duration at treatment does not predict treatment response. Could the authors reconcile the inconsistency?
Author Response
We are very grateful for these suggestions.
In answer to the specific points:
- In Figure 2, 15 exons are shown for COQ8A gene. However, in line 106-107, the authors mentioned that the COQ8A gene has 18 exons. Please clarify. --> "18" was a misprint (now corrected)
- The authors should explain the biochemical and molecular functions of the motifs listed in Figure 2, such as KxGQ, AAAS etc. This is particularly important for the discussion in line 145-150. The authors should also give some discussions on the speculations or evidence on the differential clinical severity of different missense mutations. --> we have added a more extensive discussion about that: "The protein kinase-like (PKL) domain of COQ8A protein, as for the other UbiB family proteins, possesses an N lobe insert (NLI) and a N-terminal extension (NTE) with specific motifs, illustrated in Figure 2 [8]. UbiB family proteins also have an atypical alanine-rich loop (AAAS motif) replacing the canonical glycine-rich loop in the nucleotide binding domain, an ExD motif in the NLI and a modified catalytic loop (Figure 2) [8]. It has been shown that the AAAs motif in the nucleotide-binding loop determines coenzyme selectivity and inhibits COQ8A autophosphorylation [8]. KxGQ motif in NTE domain plays a central role in inhibiting in cis autophosphorylation, but also support COQ8A ATPase activity small-molecule kinase activity (e.g., a CoQ intermediate or cofactor), as suggested by the unorthodox PKL model proposed by Stefely et al. in 2016 [7]. ... More than 120 COQ8A-related PCoQD patients and 75 pathogenic variants (including missense and loss-of-function) in COQ8A have been reported so far. COQ8A-ataxia is inherited in an autosomal recessive manner and the first case was reported in 2008 [10]. The pathogenic and likely pathogenic variants in the COQ8A gene associated with an ataxic phenotype are illustrated in Figure 2. ... Regarding missense variants, we may speculate that these genotype-phenotype correlation reflect the functional difference of AAAs motif and KxGQ domain, the former having a role in inhibiting protein autophosphorylation, the latter being essential for COQ8A role in complex Q stabilization through unorthodox PKL activity [7,8]."
- In discussing the mutations in COQ4, COQ2 and COQ5, the authors should compare the clinical severity among them as well as to the severity of COQ8A? Are different mutations associated with different physiological levels of Coenzyme Q10? Could the authors provide a brief illustrative figure on the biosynthetic pathway for coenzyme Q10 and how these various enzymes fit in the pathway? The authors then could have some discussions to synthesize the relationship between various Coenzyme Q10 related mutations and clinical severity. --> we have added a more extensive discussion about that: "Cerebellar ataxia is one of the phenotypes which have been described in PCoQD. To date, mutations in four different genes belonging to the COQ10 biosynthetic pathway have been described in PCoQD-related ataxia phenotypes, with COQ8A mutation clearly representing the vast majority (Figure 3). Cerebellar ataxia manifestation in PCoQD is typically characterized by infantile or childhood onset, whereas adult-onset ataxia have been less frequently reported. It can be present in isolation or in association with dystonia, especially for COQ8A variants. However, in the vast majority of cases it is associated with other neurological manifestations such as epilepsy, encephalopathy, spasticity, dystonia, neurodevelopmental disorder, cognitive impairment, myopathy or exercise intolerance both for COQ8A- and COQ4-related disease. A progressive childhood onset spastic-ataxia phenotype associated with mild cognitive impairment has been reported in patients harbouring COQ4 mutations, whereas COQ2 mutation have been linked to MSA-like (p or c) phenotypes. To date, a single report of COQ5-related isolated PCoQD has been reported. It is important to underlie that mutations in COQ4 and COQ2 have been linked to other different PCoQD phenotypes, such as severe infantile multisystemic form, retinopathy, hypertrophic cardiomyopathy, and steroid-resistant nephrotic syndrome. PCoQD-related ataxia diagnosis is principally driven by molecular genetic testing. Coenzyme Q10 level measurement, traditionally used to diagnose PCoQD, is now limited to confirm genetic testing results or to explore suspected cases in which molecular testing fails (i.e. presence of a variant of unknown significance in COQ genes). Regarding biochemical testing, the main evidence of PCoQD are: 1) reduced CoQ10 levels in skeletal muscle and 2) decreased activities of complex I+III and II+III in frozen muscle samples [48-50]. Alternatively, coenzyme Q10 level measurement can be performed in skin fibroblast or blood mononuclear cells, whereas plasma level is not useful because it reflects dietary intake [48-49]. Furthermore, there are cases of PCoQD with normal muscle and fibroblast Coenzyme Q10 levels, as reported by Mero et al. [22]." Furthermore, a figure about the biosynthetic pathway for coenzyme Q10 has been added (new Figure 3).
- The authors mentioned in the introduction as well as in conclusion that early diagnosis is key because it promotes early coenzyme Q10 supplementation. However, in line 255-257, the authors used reference [10] to suggest that disease duration at treatment does not predict treatment response. Could the authors reconcile the inconsistency? --> we have better specified that "response to Q10 supplementation may be incomplete or ineffective, and large studies are needed to dissect this phenomenon"
Reviewer 3 Report
Comments and Suggestions for Authors
This review is well written.
Fig1. can be labelled and explained.
It would be helpful to the reader to have explanation as to why serum coenzyme Q10 determination is not useful.
More references needed throughout the article. for example, a reference can be included for line 58 and 59.
Author Response
We are very grateful for these useful suggestions. In answer to them:
This review is well written. Fig1. can be labelled and explained. --> A legend is included
It would be helpful to the reader to have explanation as to why serum coenzyme Q10 determination is not useful. --> We explained better that coenzyme "plasma level is not useful because it reflects dietary intake" more than its production, with appropriate references
More references needed throughout the article. for example, a reference can be included for line 58 and 59. --> Done, several new references have been added
Furthermore, English language has been checked and revised (changes highlighted)
Round 2
Reviewer 2 Report
Comments and Suggestions for Authors
I appreciate the authors' editting. The authors did a great job addressing all my previous concerns. The new figure 3 is nicely presented. However, stemming from that, I have a minor point to add. Is there some current understanding on why mutations in other CoQ enzymes in the pathway did not show significant association with ataxia? The authors could add the discussion here if there is current theory and evidence in the field. If no, there is no need to speculate without much evidences.
Author Response
"I have a minor point to add. Is there some current understanding on why mutations in other CoQ enzymes in the pathway did not show significant association with ataxia? The authors could add the discussion here if there is current theory and evidence in the field. If no, there is no need to speculate without much evidences."
--> As correctly supposed by the reviewer, there is no evidence to explain why mutations in other CoQ enzymes in the pathway did not show significant association with ataxia. FOr this reason, we have added the following conclusive remark (highlighted in green): "The metabolism and functions of coenzyme Q10 are more complex than usually thought (see Figure 3). Further studies are strongly needed to unravel such complexity and to explain why mutations in some enzymes in the pathway do not show significant association with cerebellar ataxia."